# Variable Dimensionality of Europium(III) and Terbium(III) Coordination Compounds with a Flexible Hexacarboxylate Ligand

**DOI:** 10.3390/molecules27227849

**Published:** 2022-11-14

**Authors:** Xiaolin Yu, Dmitry I. Pavlov, Alexey A. Ryadun, Andrei S. Potapov, Vladimir P. Fedin

**Affiliations:** 1Nikolaev Institute of Inorganic Chemistry, Siberian Branch of the Russian Academy of Sciences, 3 Lavrentiev Ave., 630090 Novosibirsk, Russia; 2Department of Natural Sciences, Novosibirsk State University, 2 Pirogov Str., 630090 Novosibirsk, Russia

**Keywords:** europium, terbium, flexible ligands, metal–organic frameworks, coordination polymers, luminescence, crystal structure

## Abstract

A reaction between 4,4′,4″-(benzene-1,3,5-triyltris(oxy))triphthalic acid (H_6_L) and lanthanide(III) nitrates (Ln = Eu^3+^, Tb^3+^) in water under the same conditions gave a molecular coordination compound [Tb(H_4.5_L)_2_(H_2_O)_5_]∙6H_2_O in the case of terbium(III) and a one-dimensional linear coordination polymer {[Eu_2_(H_3_L)_2_(H_2_O)_6_]∙8H_2_O}_n_ in the case of europium(III). The crystal structures of both compounds were established by single-crystal X-ray diffraction, and they were further characterized by powder X-ray diffraction, thermogravimetric analysis and infrared spectroscopy. The compounds demonstrated characteristic lanthanide-centered photoluminescence. The lanthanide-dependent dimensionality of the synthesized compounds, which are the first examples of the coordination compounds of hexacarboxylic acid H_6_L demonstrates its potential as a linker for new coordination polymers.

## 1. Introduction

Aromatic polycarboxylic acids are important building blocks for the construction of metal–organic frameworks (MOFs), since the carboxylate groups are able to form strong coordination bonds with most metal cations [1,2,3,4]. Rigid di-, tri- and tetracarboxylic acids were successfully used for the preparation of robust MOFs [5,6,7]. In recent years, an increasing amount of attention has been paid to flexible MOFs, which are able to expand or contract their networks upon external stimuli, such as temperature, pressure or electromagnetic irradiation [8,9,10,11,12,13]. The flexibility of MOFs can be induced by the conformational flexibility of their organic linkers; the latter may be achieved by introducing ether groups into their structures [14,15,16,17]. Increasing the number of the carboxylic groups in MOF linkers may leave some of them uncoordinated by the metal nodes and improve the functional properties of the resulting MOF, such as proton conductivity, adsorption selectivity for gases from their mixtures or metal ions from solutions [18].

In this work, we explore the coordination chemistry of an ether-bridged hexa-carboxylate ligand, 4,4’,4’’-(benzene-1,3,5-triyltris(oxy))triphthalic acid (H_6_L, Figure 1), for which no coordination compounds have been reported before, although the compound itself was first prepared two decades ago and used for the synthesis of hyperbranched organic polymers [19]. Two lanthanide metals, europium(III) and terbium(III), were chosen as the central ions for the synthesis of coordination compounds because of their potential luminescent properties [20,21,22,23,24,25,26,27], possible applications as catalytic platforms [28,29], proton-conductive or magnetic materials [30]. In addition, flexible polycarboxylate ligands are especially suitable for the formation of lanthanide coordination polymers because of their ability to form strong metal-oxygen bonds and saturate the high lanthanide coordination numbers [31,32].

## 2. Results and Discussion

### 2.1. Synthesis of the Coordination Compounds

Terbium(III) and europium(III) coordination compounds were prepared as single crystals by the reaction between H_6_L and metal nitrate hexahydrates (1:1 molar ratio) in an aqueous solution under hydrothermal conditions (Figure 1). It was found that specific pH conditions are necessary for the formation of the products. Thus, 3 equivalents (relative to the amount of the metal salt) of the base (KOH) were necessary to carry out the reaction at 90 °C, followed by neutralization with HNO_3_ (2.5 equivalents) and prolonged crystallization at room temperature. Numerous attempts to carry out the reaction under other conditions (no acid of base added, only acid or base added, variable temperature) did not result in the formation of any solid product. When the amounts of acid or base used were different from the ones indicated above, only amorphous solids were obtained that were not further characterized. It is interesting to note that under identical synthetic conditions, Tb^3+^ formed a molecular complex with a 1:2 M:L ratio, while Eu^3+^ gave a linear coordination polymer of a 1:1 M:L composition. Other M:L ratios from 6:1 to 1:6 were tested as well, but they gave no solid products or only amorphous precipitates.

### 2.2. Crystal Structure of the Coordination Compounds

Single-crystal X-ray diffraction analysis revealed that compound **1** crystallizes in a monoclinic crystal system, space group C2/c. The compound is a molecular complex, consisting of one Tb^3+^ ion, five coordinated water molecules, two crystallographically equivalent ligand molecules and six lattice water molecules (Figure 1a). Each Tb^3+^ ion coordinates nine oxygen atoms, four of which are from the carboxylic groups of two organic ligands and the remaining five are from the coordinated water molecules. According to the Shape 2.1 software package [33], the coordination environment is best described by a muffin configuration (MFF) [34] (Figure 1b, Appendix A). The Tb-O distances are 2.397(2) and 2.547(2) Å for the carboxylate ligand and vary from 2.330(2) Å to 2.418(2) Å for the coordinated water molecules, which are typical values for such types of coordination. The proton of the carboxylic group at the *ortho*-position to the coordinated carboxylate group participates in an intramolecular hydrogen bond (d(D···A) = 2.394(3) Å, d(H···A) = 1.25(4) Å, d(D–H) = 1.14(4) Å, ∠(D–H···A) = 174(3)°) and is disordered over two equivalent positions in two anionic ligands (Appendix A). To achieve electroneutrality, the terbium(III) coordination sphere should therefore be composed as [Tb(H_4.5_L)_2_(H_2_O)_5_]. The benzene rings in compound **1** participate in intermolecular π-π stacking interactions with the centroid-to-centroid separation of 3.791 Å (the angle between the benzene ring planes is 10.9°, Appendix A), and the water molecules are involved in multipoint hydrogen bonds (Appendix A), which connect the molecules of compound **1** into a three-dimensional supramolecular framework (Figure 1c).

Although the Eu-compound **2** was synthesized by the same method as the Tb-compound **1**, the single crystal X-ray diffraction analysis of compound **2** reveals a completely different crystal structure. Compound **2** crystallizes in a triclinic crystal system with space group P-1; the asymmetric unit contains one Eu^3+^ ion, one H_3_L^3-^ ligand, three coordinated water molecules and four lattice water molecules (Figure 2a). Similarly to compound **1**, the coordination polyhedron of Eu^3+^ is close to the muffin shape (Appendix A), but in contrast to the Tb-compound **1**, two Eu^3+^ cations are joined by the bridging carboxylate groups into binuclear 8-connected [Eu_2_] secondary building units (Figure 2b), which are connected by two H_3_L^3-^ linkers into 1D infinite chains parallel to the *ac*-plane (Figure 2c). The Eu–O distances for the coordinated carboxylate groups are 2.385(2) Å and 2.526(2) Å, typical for europium(III) carboxylate compounds. Further, a 3D supramolecular structure is formed between one-dimensional chains through the hydrogen bonding interactions, involving the coordinated carboxylate and the uncoordinated carboxylic groups (Appendix A, Figure 2d and Appendix A).

Different coordination behaviors of Tb^3+^ and Eu^3+^ towards H_6_L ligand under the same conditions may be attributed to the known lanthanide contraction effect. Despite a weak monotonic change of the ionic radii in the lanthanide series, for a certain Ln^3+^ ion, a structure may cease to be stable due to the increased steric hindrance in the ever-shrinking coordination sphere of the metal ion. This can lead to a change in the ligand connectivity, including a change in dimensionality and topology. Gadolinium, which stands between the europium and terbium in the lanthanide series, often appears to be a breaking point in such alternations and one product is formed for lanthanides lighter than Gd and a different product is obtained for heavier lanthanides [35,36,37].

The conformational flexibility of the potentially hexacarboxylate ligand H_6_L is evident from the variability of the dihedral angles between the benzene rings connected by ether bonds in Tb-compound **1** and Eu-compound **2** from 61.5° to 88.8° (Figure 3).

### 2.3. X-ray Powder Diffraction, IR Spectroscopy and Thermogravimetric Analysys

The phase purity of compounds **1** and **2** was confirmed by powder X-ray diffraction. As shown in Appendix A, the experimental diffractograms recorded at room temperature and the simulated patterns obtained from single crystal data are in good agreement.

The IR spectra of the H_6_L ligand, Tb-compound **1** and Eu-compound **2** are shown in Appendix A. The IR spectrum of Tb-compound **1** features an absorption peak near 3037 cm^−1^, which is attributed to the C–H stretching vibrations of the aromatic ring. The benzene ring vibration bands were observed at 1582 cm^−1^ and 1549 cm^−1^. In addition, the spectrum of Tb-compound **1** demonstrates a wide O-H stretching vibration peak near 3390 cm^−1^ from water molecules and protonated carboxylate groups involved in hydrogen bonding. The characteristic peaks at 1737 cm^−1^ and 1711 cm^−1^ may be assigned to the stretching vibrations of the protonated carboxyl groups, in accordance with incomplete deprotonation of the H_6_L ligand. The strong bands at 1582 cm^−1^, 1549 cm^−1^, 1455 cm^−1^ and 1425 cm^−1^ correspond to the asymmetric and symmetric stretching vibrations of the carboxylate groups. The IR spectrum of Eu-compound **2** demonstrated features similar to the spectrum of Tb-compound **1**. Thus, the peaks near 3080 cm^−1^ and 3390 cm^−1^ were attributed to C–H and O–H stretching vibrations, and correspondingly, the benzene ring vibrations were observed at 1596 cm^−1^ and 1431 cm^−1^. The band at 1721 cm^−1^ corresponds to the C = O carbonyl stretching of COOH groups, while the bands near 1596 cm^−1^, 1554 cm^−1^ and 1431 cm^−1^ were assigned to the asymmetric and symmetric stretching vibrations of the carboxylate groups.

The TG curve of Tb-compound **1** exhibited a weight loss of 14.3%, which occurred in two steps within 30–200 °C (Appendix A). The first step in the range of 30–150 °C is due to the removal of free water molecules (found: 7.4%, calc.: 6.8%), and the second step at 150–200 °C is due to the removal of coordinated water molecules (found: 6.9%, calc.: 5.7%). After dehydration, Tb-compound **1** remains stable up to 360 °C, indicative of good thermal stability.

### 2.4. Luminescent Properties of the Ligand and the Coordinaion Compounds

The solid-state luminescent spectra of H_6_L ligand, Tb-compound **1** and Eu-compound **2** were measured for the powdered samples at room temperature. As shown in Appendix A, the H_6_L ligand showed a broad emission peak with a maximum of 456 nm (λ_ex_ = 370 nm). The excitation spectra of both Tb-compound **1** and Eu-compound **2** showed broad bands with maxima near 300 nm (Appendix A), suggesting ligand-centered absorption. When excited at 300 nm, the Tb-compound **1** exhibited the following four characteristic emission peaks from the Tb^3+^ ion: 490, 545, 584 and 622 nm, attributed to ^5^D_4_→^7^F_6_, ^5^D_4_→^7^F_5,_ ^5^D_4_→^7^F_4_, ^5^D_4_→^7^F_3_ transitions [38], respectively (Figure 4a). Likewise, upon excitation at 310 nm, the Eu-compound **2** exhibited the following four characteristic emission peaks from the Eu^3+^ ion: 593, 615, 649 and 694 nm, attributed to ^5^D_0_→^7^F_1_, ^5^D_0_→^7^F_2_, ^5^D_0_→^7^F_3_, ^5^D_0_→^7^F_4_ transitions [39], respectively (Figure 4b). The luminescence lifetime of the Tb-compound **1** obeys a single exponential equation (characteristic lifetime of 0.68 ms), which indicates that a single coordination environment exists for Tb^3+^ ion (Figure 4c). The luminescence lifetime of the Eu-compound **2** was 0.28 ms (Figure 4c). The quantum yields of the Tb-compound **1** and the Eu-compound **2** were 8% and 2%, respectively. Relatively low quantum yields may be due to the conformational flexibility of the ligand, leading to vibrational non-radiative energy dissipation, often observed for lanthanide coordination polymers with flexible ligands [40,41]. In addition, the presence of five or three coordinated water molecules in the lanthanide coordination sphere in compounds **1** and **2** also leads to deactivation through O–H vibrations [42].

As shown in Figure 4d, the emission of compounds **1** and **2** is characterized by the chromaticity coordinates (0.3318, 0.5769) and (0.6425, 0.3464). The color temperature of the green emission of Tb-compound **1** was 5600 K, and the red emission of the Eu-compound **2** had a color temperature of 8870 K, both of which correspond to the cool colors (>5000 K are called cool colors).

In order to gain insight into the photoluminescence mechanism, TD-DFT calculations for the H_6_L ligand and its triply deprotonated form (as lithium salt, Li_3_H_3_L) were carried out. In the optimized structure of the Li_3_H_3_L model, the dihedral angles corresponding to the rotation of the phthalate rings are in good agreement with the values obtained from the X-ray crystal structure of compound **2**, suggesting that the predicted conformation of the anionic ligand is approximately the same as in the structure of compound **2** (Appendix A, Appendix A). Therefore, the obtained geometry of Li_3_H_3_L may be used for further calculations of the photophysical properties. According to TD-DFT calculations, the UV-Vis absorption of H_6_L is associated with the S_0_→S_1_ excitation; the calculated maximum is 302 nm. The S_0_→S_1_ excitation is accompanied by the following three major electron transitions: HOMO→LUMO (contribution 49%), HOMO-1→LUMO (contribution 41%), HOMO-2→LUMO (contribution 10%). As one can see from the localization of the molecular orbitals, the S_0_→S_1_ excitation is a π→π* transition with the charge transfer between the aromatic rings of H_6_L (Appendix A).

The calculated position of the absorption maximum of Li_3_H_3_L is 316 nm, which is in reasonable agreement with the experimentally observed value of 308 nm for compound 2. The S_0_→S_1_ excitation is characterized by the following two major electronic transitions: HOMO→LUMO (68%) and HOMO-1→LUMO (32%). The charge transfer accompanying the π→π* excitation in Li_3_H_3_L is even more pronounced compared to the protonated ligand H_6_L (Figure 5).

Relatively long luminescence lifetimes of the coordination compounds **1** and **2** suggest an emission due to f-f lanthanide transitions. At the same time, the broad absorption bands near 310 nm suggest a ligand-centered excitation; therefore, an intersystem crossing S_1_-T_1_ process must be assumed, followed by an energy transfer from the T_1_ state to ^5^D_0_ or ^5^D_4_ states of Eu^3+^ or Tb^3+^ ions. A comparison of the energies of these states indicates that in both cases, such transitions are energetically favorable (Figure 6).

## 3. Materials and Methods

### 3.1. Starting Materials and Synthetic Procedures

All reagents were commercially available and used without further purification. The ligand 4,4′,4″-(benzene-1,3,5-triyltris(oxy))triphthalic acid (H_6_L) was obtained from Jinan Henghua Sci. & Tec. Co. Ltd. (Jinan, China) and used as received.

#### 3.1.1. Synthesis of Compound [Tb(H_4.5_L)_2_(H_2_O)_5_]∙6H_2_O (**1**)

A mixture containing Tb(NO_3_)_3_∙6H_2_O (3.6 mg, 0.008 mmol), H_6_L (5 mg, 0.008 mmol) were dissolved in H_2_O (2 mL) and then 24 μL of 1 M KOH solution was added. The solution was sealed in a screw-cap vial and heated at 90 °C for 48 h, gradually cooled to room temperature and 20 μL of 1 M HNO_3_ solution were added. The resulting clear solution was heated at 100 °C for 24 h, gradually cooled to room temperature again and allowed to stand for 2 weeks to obtain pale yellow flaky crystals. Yield: 47% (based on H_6_L). Elemental analysis calcd. (%) for C_60_H_55_O_41_Tb: C, 45.3; H, 3.5. Found (%) C, 45.3; H, 3.2. IR (cm^−1^, KBr): 3390 (m), 3073 (m), 2929 (m), 2612 (w), 2144 (w), 1737 (m), 1711 (m), 1662 (m), 1582 (s), 1549 (s), 1455 (s), 1425 (s), 1374 (s), 1329 (m), 1307 (m), 1264 (m), 1223 (s), 1128 (m), 1064 (m), 1010 (m), 924 (w), 883 (w), 857 (w), 795 (w), 774 (w), 743 (w), 711 (w), 670 (w), 638 (w), 533 (w), 473 (w), 439 (w).

#### 3.1.2. Synthesis of Compound {[Eu_2_(H_3_L)_2_(H_2_O)_6_]∙8H_2_O}_n_ (**2**)

Compound **2** was synthesized in the same way as compound **1**, except that Tb(NO_3_)_3_∙6H_2_O was replaced by Eu(NO_3_)_3_∙6H_2_O (3.6 mg, 0.008 mmol). Yield: 52% (based on H_6_L). Elemental analysis calcd. (%) for C_60_H_50_Eu_2_O_40_: C, 42.0; H, 2.9. Found (%) C, 42.2; H, 2.9. IR (cm^−1^, KBr): 3386 (m), 3082 (m), 2925 (w), 1721 (m), 1674 (m), 1596 (s), 1554 (s), 1431 (s), 1385 (s), 1322 (m), 1267 (m), 1221 (s), 1142 (m), 1118 (m), 1069 (w), 1008 (m), 922 (w), 850 (w), 836 (w), 821 (w), 804 (w), 781 (w), 710 (w), 680 (w), 656 (w), 637 (w), 610 (w), 572 (w), 458 (w).

### 3.2. Physical Methods of Analysis

Powder X-ray diffraction (PXRD) measurements were carried out using a Bruker D8 ADVANCE diffractometer (Bruker Corporation, Billerica, MA, USA), Cu-Kα radiation, λ = 1.5406 Å, 2θ range 3–40°. Elemental analyses for C and H were performed using a Vario MICRO Cube analyzer (Elementar Analysensysteme GmbH, Langenselbold, Germany). Thermogravimetric analysis (TGA) was carried on a NETZSCH TG 209 F1 Iris Thermo Microbalance (Erich NETZSCH GmbH & Co. Holding KG, Selb, Germany) heated from 30 to 850 °C under helium atmosphere with the heating rate of 10 °C/min. IR spectra were recorded on a Bruker Scimitar FTS 2000 spectrometer (Bruker Corporation, Billerica, MA, USA) in the range 4000–400 cm^−1^. Luminescence spectra, luminescence lifetimes and quantum yields were obtained on Horiba Jobin Yvon Fluorolog 3 (HORIBA Jobin Yvon SAS, Edison, NJ, USA) photoluminescence spectrometer equipped with 450W ozone-free Xe-lamp, a cooled photon detection module and an integrating sphere.

### 3.3. Single-Crystal X-ray Diffraction

Diffraction data for single crystals compounds **1** and **2** were collected with a Bruker D8 Venture diffractometer with a CMOS PHOTON III detector and IμS 3.0 source (mirror optics, λ(CuKα) = 1.54178 Å). The φ-and ω-scanning techniques were employed to measure the intensities. The crystal structures were solved and refined by means of the SHELXT [43] and SHELXL [44] programs using OLEX2 GUI [45]. Atomic displacement parameters for non-hydrogen atoms were refined anisotropically. Hydrogen atoms were placed geometrically and refined in the riding model. The crystallographic parameters and the details of the diffraction experiment are given in Table 1. The bond lengths and bond angles for the Tb-compound **1** and Eu-compound **2** are provided in Appendix A.

### 3.4. Computational Chemistry Details

The calculations were performed using Gaussian 09 package [46]. The isolated H_6_L molecule was used to model the free ligand, while a triply deprotonated form (balanced by three lithium cations, Li_3_H_3_L) was used to represent the anionic ligand in compound **2**. Singlet ground state geometry optimizations of H_6_L were carried out in the gas phase at the DFT level of theory employing the three-parameter hybrid B3LYP functional [47,48,49,50] and 6–31 + G(d) basis set [51,52,53,54]. An empirical dispersion correction was applied using the D3 version of Grimme’s empirical dispersion with Becke-Johnson damping [55]. The frequency calculations in a harmonic approximation were performed for the optimized geometries in order to establish the nature of the stationary points, lack of imaginary vibration modes for the optimized structures indicates that the stationary points found corresponded to minima on the potential energy surface. The first singlet and triplet exited states of H_6_L and Li_3_H_3_L were computed at time-dependent DFT (TD-DFT) level, using the optimized ground state geometry and the same functional and basis set used for the ground state calculations. 

## 4. Conclusions

In summary, the coordination chemistry of a flexible aromatic triether-bridged hexacarboxylate ligand (4,4′,4″-(benzene-1,3,5-triyltris(oxy))triphthalic acid) was studied for the first time, and two new lanthanide coordination compounds were prepared and characterized. It was found that the nature of the lanthanide affects the dimensionality of the compounds formed. Thus, in identical reaction conditions, Tb^3+^ forms a discrete coordination compound, while Eu^3+^ yields a 1D coordination polymer. Both compounds demonstrated characteristic lanthanide-centered emission and ligand-centered excitation, in accordance with the experimental absorption spectra and TD-DFT calculations. Therefore, the aromatic ligand acts as an antenna for lanthanide excitation and further studies of its coordination chemistry may lead to the preparation of efficient light emitters. It should also be noted that the observed difference in the reactivity of Eu^3+^ and Tb^3+^ may contribute to solving the problem of lanthanide separation.

## Data Availability

Experimental data associated with this research are available from the authors. Crystallographic data for the structural analysis were deposited at the Cambridge Crystallographic Data Centre, CCDC No. 2214908 for compound **1**, 2214910 for compound **2**. Copies of the data can be obtained free of charge from the Cambridge Crystallographic Data Centre, 12 Union Road, Cambridge CB2 1EZ, UK (fax: +44-1223-336-033; e-mail:deposit@ccdc.cam.ac.uk).

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
