# Peer review of "Variable Dimensionality of Europium(III) and Terbium(III) Coordination Compounds with a Flexible Hexacarboxylate Ligand"

_molecules, 2022, doi:10.3390/molecules27227849_

Round 1
Reviewer 1 Report
Sentence, A reaction between 4,4',4''-(benzene-1,3,5-triyltris(oxy))triphthalic acid (H6L) and ter- 10 bium(III) .........in the case of europium(III)” in abstract need to be modified.
Revise the sentence, “Aromatic polycarboxylic acids........ most metal cations [1–3]”
Following paper can be cited in this section
Dalton Trans.2017,46, 13943-13951
Following references can be added in the text “In the recent years......
electromagnetic irradi- 29 ation [7–11].
Dalton Trans., 2017,46, 9022-9029
trieher-hexacarbocylate ligand?
Synthesis of the coordination compounds: Authors must specify how much of the starting materials were taken and what was the pH during the whole reaction.
monoclinic space group C2/c?
Check the sentence the” asymmetric unit of contains one Eu3+ ions, one H3L3- ligand, three coordinated water mol- 89 ecules and four lattice water molecules”
Authors must explain various weak interactions in both compounds
Referencs need to be double checked. I didn’t see references
Authors must check the language throughout the manuscript. There are several badly constructed sentences
Author Response
Reviewer 1
Sentence, A reaction between 4,4',4''-(benzene-1,3,5-triyltris(oxy))triphthalic acid (H6L) and terbium(III).in the case of europium(III)” in abstract need to be modified.
The sentence was revised to avoid repeating the names of the elements.
Revise the sentence, “Aromatic polycarboxylic acids. most metal cations [1–3]” Following paper can be cited in this section Dalton Trans.2017,46, 13943-13951
The suggested reference was cited in this section.
Following references can be added in the text “In the recent years......electromagnetic irradiation [7–11].” Dalton Trans., 2017,46, 9022-9029
The suggested reference was cited in this section.
trieher-hexacarbocylate ligand ?
the general name of the ligand was changed to “trieher-bridged hexa-carboxylate ligand”
Synthesis of the coordination compounds: Authors must specify how much of the starting materials were taken and what was the pH during the whole reaction.
The quantities of the starting materials are already given in the experimental procedures (sections 3.1.1. and 3.1.2.). The pH was not monitored, however, the exact amounts of KOH or HNO3 added at different stages of the reactions are given in the experimental procedures to ensure the reproducibility.
monoclinic space group C2/c ?
This phrase was revised to “monoclinic crystal system, space group C2/c”
Check the sentence the” asymmetric unit of contains one Eu3+ ions, one H3L3- ligand, three coordinated water molecules and four lattice water molecules”
The phrase was revised, “of” was removed
Authors must explain various weak interactions in both compounds
The discussion of the intermolecular interactions in lines 79-83 and 98-101 were appended by some geometrical parameters and two additional figures were placed in the supplementary materials to get a better representation of these interactions (Figures S2 and S3).
References need to be double checked. I didn’t see references
The reference list is checked and was included in the original manuscript.
Authors must check the language throughout the manuscript. There are several badly constructed sentences
The text was proof-read and polished, where necessary.
Reviewer 2 Report
The article “Variable dimensionality of europium(III) and terbium(III) coordination compounds with a flexible hexacarboxylate ligand” by V. P. Fedin” and co-authors reports an investigation of 4,4’,4’’-(benzene-1,3,5-triyltris(oxy))triphthalic acid (H6L, Ligand) as a polytopic ligand for construction of flexible Metal Organic Frameworks (MOF) based on example of H6L reaction with terbium(III) or europium(III) nitrate in water under hydrothermal conditions.
The crystal structures of both complex compounds were established by single-crystal XRD analysis and they were further characterized by powder XRD analysis, thermogravimetric analysis and infrared spectroscopy. Photophysical properties of both Ln(III) complexes were studied on simple level.
The article reports studies that are interesting as an expanding of family of flexible bridging ligands for MOF and their complexes with lanthanides, and can be recommend it for publication in Molecules, issue “Synthesis, Characterization and Crystal Structure of Coordination Compounds” after some revision that are rather minor.
The article is very good organized, written in good and transparent English and there is no problem for a reader to extract the main idea and to understand details.
The following points should be answered:
(1) Authors shown the reactivity difference of the Ligand toward Eu(III) and Tb(III) but did not try to explain this phenomenon.
(2) Why only 1:1 Ln(III):Ligand molar ration has been tested? The Ligand contains six donor sites and 6:1 Ln(III):Ligand molar ration should demonstrate the real potential of the Ligand for MOF construction.
(3) As it is shown on Scheme 1, compound 1 is not electrostatically neutral, the sum of charges of Tb(III) and two H5L− is not zero. Please, comment it is the text or show counterion.
(4) Calculated and experimental PXRD patterns for Eu(III) complex are not similar, especially in diapason of large 2theta angles (Figure S2).
(5) Caption of Figure 3. Please, check the subscripts and superscripts for H3L3.
(6) CHN elemental analysis should be given for 1 and 2.
(7) Luminescent properties of the Ligand and Ln(III) complexes:
(a) Please, double check f-f transitions for complexes 1 and 2 (main text, Figures 4a,b and 6). The reviews DOI: 10.1016/j.ccr.2015.02.015 and DOI: 10.1021/cr400477t can be recommended as example.
(b) Luminescence lifetime and quantum yield for the both Ln(III) complexes are relatively low. Please, add in the text a comparison with similar literature examples and evaluate the role of coordinated water in f-f luminescent quenching.
(c) Please, show curve of the afterglow time for 1 and 2 (Figure 4c,d) in the same scale to demonstrate the difference in lanthanide properties. They can be placed on the same picture.
(d) CIE for 1 and 2 emission can be placed on the same picture.
(e) Please, attribute f-f transitions in solid-state excitation spectra of 2 (Figure S6b).
(f) DFT calculation is not the best way to indicate T1 level (or 0→0 transition) for the Ligand in compounds 1 and 2 because experiment is performed for deprotonated Ligand and in the solid, but calculations are performed for fully protonated Ligand (means the other distribution of electrostatic potential) and in gaseous phase.
(g) Triplet energy of the Ligand in 1 and 2 has to be found by experimental way through low temperature emission of Gd(III) complex (see, for example DOI: 10.1039/c8qi00712h).
(h) The luminescence of 1 and 2 is not phosphorescence-type emission, it is emission due to intrametal f-f transitions and are possible for complexes due to antenna effect.
(8) Carefully check the text of the Conclusion, it looks like Eu(III) and Tb(III) have been misplaced.
Author Response
Reviewer 2
The article “Variable dimensionality of europium(III) and terbium(III) coordination compounds with a flexible hexacarboxylate ligand” by V. P. Fedin” and co-authors reports an investigation of 4,4’,4’’-(benzene-1,3,5-triyltris(oxy))triphthalic acid (H6L, Ligand) as a polytopic ligand for construction of flexible Metal Organic Frameworks (MOF) based on example of H6L reaction with terbium(III) or europium(III) nitrate in water under hydrothermal conditions.
The crystal structures of both complex compounds were established by single-crystal XRD analysis and they were further characterized by powder XRD analysis, thermogravimetric analysis and infrared spectroscopy. Photophysical properties of both Ln(III) complexes were studied on simple level.
The article reports studies that are interesting as an expanding of family of flexible bridging ligands for MOF and their complexes with lanthanides, and can be recommend it for publication in Molecules, issue “Synthesis, Characterization and Crystal Structure of Coordination Compounds” after some revision that are rather minor.
The article is very good organized, written in good and transparent English and there is no problem for a reader to extract the main idea and to understand details.
The following points should be answered:
(1) Authors shown the reactivity difference of the Ligand toward Eu(III) and Tb(III) but did not try to explain this phenomenon.
A discussion was added in lines 105-113.
(2) Why only 1:1 Ln(III):Ligand molar ration has been tested? The Ligand contains six donor sites and 6:1 Ln(III):Ligand molar ration should demonstrate the real potential of the Ligand for MOF construction.
Different M:L ratios were indeed tested, but no characterizable products were formed, a note on this fact was added to the manuscript, lines 61-62.
(3) As it is shown on Scheme 1, compound 1 is not electrostatically neutral, the sum of charges of Tb(III) and two H5L− is not zero. Please, comment it is the text or show counterion.
The formula of the compound is [Tb(H4.5L)2(H2O)5]∙6H2O, it is the same throughout the manuscript. One carboxylic proton occupies two positions in two COOH groups with equal fractions. This was noted in lines 77-78. To avoid misunderstanding, an additional note was added to the Figure 1 caption.
(4) Calculated and experimental PXRD patterns for Eu(III) complex are not similar, especially in diapason of large 2theta angles (Figure S2).
Eu(III) complex was obtained as a uniform crystalline precipitate (see inset in Figure S2). In the characteristic low 2theta region there is a good agreement between the calculated and experimental PXRD patterns. In higher 2theta region both the calculated and experimental PXRD patterns contain a lot of reflexes, which makes the visual comparison difficult in this region. The difference in the intensities of the calculated and the experimental reflexes is possibly due to the preferred orientation of microcrystals in the powder. Taking into account the potential flexibility of the structure of compound 2 (due to three ether bridges in the ligand), different measurement conditions (150 K for single crystals and 298 K for powder), may lead to differences in the diffraction patterns in the higher 2theta region.
(5) Caption of Figure 3. Please, check the subscripts and superscripts for H3L3.
Corrected
(6) CHN elemental analysis should be given for 1 and 2.
The elemental analysis results were presented in the original manuscript in the experimental procedures (lines 221-222 and 229-230).
(7) Luminescent properties of the Ligand and Ln(III) complexes:
(a) Please, double check f-f transitions for complexes 1 and 2 (main text, Figures 4a,b and 6). The reviews DOI: 10.1016/j.ccr.2015.02.015 and DOI: 10.1021/cr400477t can be recommended as example.
We have checked the transitions and found them consistent with the literature data. The mentioned relevant reviews were cited where these transitions are first mentioned.
(b) Luminescence lifetime and quantum yield for the both Ln(III) complexes are relatively low. Please, add in the text a comparison with similar literature examples and evaluate the role of coordinated water in f-f luminescent quenching.
A discussion on the possible deactivation pathways was added in lines 161-166.
(c) Please, show curve of the afterglow time for 1 and 2 (Figure 4c,d) in the same scale to demonstrate the difference in lanthanide properties. They can be placed on the same picture.
The Figure 4 was revised, panels c and d were merged.
(d) CIE for 1 and 2 emission can be placed on the same picture.
The Figure 4 was revised, panels e and f were merged.
(e) Please, attribute f-f transitions in solid-state excitation spectra of 2 (Figure S6b).
The transitions were labeled and Figure S6b was updated.
(f) DFT calculation is not the best way to indicate T1 level (or 0→0 transition) for the Ligand in compounds 1 and 2 because experiment is performed for deprotonated Ligand and in the solid, but calculations are performed for fully protonated Ligand (means the other distribution of electrostatic potential) and in gaseous phase.
The calculations were repeated for a triply deprotonated form (and adding three Li+ cations to avoid generating excessive negative charge). We believe the gas phase approximation is acceptable in our case, since only a solid-state absorption is considered and using PCM solvation models would be equally far from the experimental conditions. Solid-state ONIOM or periodic boundary conditions models would be more appropriate for this care, but for the purpose of qualitative comparison of the energy levels they seem to be excessive.
Alter taking the deprotonation into account (as in Li3H3L model), the S1 and T1 energy levels have changed only slightly and the relative positions did not change, but the Figure 6 was revised with the new values.
The computational part was rewritten, all of the related figures and tables were updated.
(g) Triplet energy of the Ligand in 1 and 2 has to be found by experimental way through low temperature emission of Gd(III) complex (see, for example DOI: 10.1039/c8qi00712h).
We agree that this would be the best approach to assess the T1 level, however, taking into account prolonged crystallization times and differing structures of the products for different Ln3+ cations, we believe it is unnecessary to attempt preparing and characterizing the Gd(III) complex in the specific case of the compounds discussed in the manuscript. We find the suggested work with in-depth analysis of the lanthanide luminescence quite useful and cite it in the revised manuscript.
(h) The luminescence of 1 and 2 is not phosphorescence-type emission, it is emission due to intrametal f-f transitions and are possible for complexes due to antenna effect.
The phrase in lines 199-201 was revised, thank you for pointing out this terminological inaccuracy.
(8) Carefully check the text of the Conclusion, it looks like Eu(III) and Tb(III) have been misplaced.
Eu and Tb were indeed mismatched, thank you for the attentive reading of the manuscript.
Round 2
Reviewer 1 Report
Authors must make a thorough knowledge before writing the abstract
Authors must write Lanthanide nitrate and in bracket must mention [Ln = Tb and Eu]
Authors uploaded the review file at the time of submission of the revision. Take care of this
Authors must check the langauge
Author Response
Reviewer 1
Authors must make a thorough knowledge before writing the abstract
We believe the abstract in its current form is concise, written in a clear language and fully represents the contents of the manuscript.
Authors must write Lanthanide nitrate and in bracket must mention [Ln = Tb and Eu]
Corrected (line 11)
Authors uploaded the review file at the time of submission of the revision. Take care of this
The revised file with tracked changes is uploaded according to the requirements of the editorial office. The newly revised file also contains the tracked changes.
Authors must check the language
The manuscript was thoroughly checked; several typographic errors were corrected and the missing articles were added.
Thank you for considering our manuscript for publication.
Yours sincerely,
Prof. Dr. Vladimir P. Fedin,
Nikolaev Institute of Inorganic Chemistry
Novosibirsk, Russia